# Disaster Risk Assessment Scheme—A Road System Survey for Budapest

Tibor Sipos [1,2], Zsombor Szabó [1,3], Mohammed Obaid [4] and Árpád Török [4,*]

1 Department of Transport Technology and Economics, Faculty of Transportation Engineering and Vehicle Engineering, Budapest University of Technology and Economics, 1111 Budapest, Hungary
2 Directorate for Research and Innovation, KTI Institute for Transport Sciences Nonprofit Ltd., 1119 Budapest, Hungary
3 Directorate for Public Transport Services, KTI Institute for Transport Sciences Nonprofit Ltd., 1119 Budapest, Hungary
4 Department of Automotive Technology, Faculty of Transportation Engineering and Vehicle Engineering, Budapest University of Technology and Economics, 1111 Budapest, Hungary
* Correspondence: torok.arpad@kjk.bme.hu

**Abstract:** This study presents a method to analyze the most critical elements of the public road system concerning outer effects which hinder the normal operation of the whole system. The surveyed public road network in Budapest, Hungary is studied as a graph: Dijkstra's algorithm is applied to find the shortest path, and the Boykov-Kolmogorov method is used to calculate the maximum flow of the network. Those elements are identified whose damage can critically influence the operation of the network, and where the infrastructure available for rescue teams has a bottleneck. Finally, the Wilcoxon post hoc test was applied with Bonferroni correction. The tests have proven that the new method can successfully identify the critical vulnerabilities of the network to determine its weak points by considering reduced road capacities and the increased needs for transportation arising due to a disaster. This pilot study confirmed that after the elimination of the problems in statistical methods, the new framework can robustly identify those road network elements whose development is of key importance from a disaster management perspective.

**Keywords:** network vulnerability; network resilience; critical event analysis; safety and security

## 1. Introduction

From the point of view of sustainability, the aspects of safety and reliability are essential. In accordance with this, the Global Sustainable Development Report also deals with the topic. The mentioned report emphasizes that the proper and efficient operation of the transport system is very important to maintain a sustainable future.

The present research aims to examine systematic outer factors that might hinder the efficient operation of the public road network, e.g., in the case of emergency situations caused by natural disasters. This study proposes a method for the identification and ranking of the most critical elements of the public road network.

For all decision makers, it is of utmost importance that the transport network of the given region or area is resistant to various natural disasters, as well as to deliberate sabotage [1–3]. If some undesirable event occurs, the prevention and rescue management must be as efficient as possible. However, the disaster that has occurred can damage the transport infrastructure network to such a negative extent that it can make the situation of both the emergency responders and those trying to leave the area difficult.

Thus, to make the topic more understandable, essential definitions regarding the topic must be clarified.

- Disaster: a set of undesirable events that occur due to some natural cause (e.g., flood, earthquake) or deliberate sabotage (explosion) and entail evacuating the population from a given area;
- Disaster management: post-disaster activities to minimize losses, including evacuating the affected population and prevention;
- Disaster risk reduction: measures to make transport infrastructure more resilient to the consequences of disaster events.

Research conducted by Suyfan, Saqib, and Zia aimed to create a methodological framework for identifying the critical elements of the communication network [4]. On the one hand, their main objective was to estimate the effects of some damage influencing the operation of the system elements; on the other hand, they also identified the so-called critical elements of the network, which primarily influence the operation of the whole network. The study by Leal, Oliveira, and Porto [5] goes beyond the examination of general network perspectives and focuses on the vulnerable elements of the public road network. The effects of catastrophes and unexpected natural phenomena are estimated by Luathep and co-authors [6].

Ferretti and his colleagues realized that the existence of huge databases describing transport needs and patterns make the improvement of transport network sensitivity possible, especially concerning the risks of catastrophic situations. Consequently, they created a new model to map the mobility needs connected to each mode of transport. This new method is suitable for exploring the changes in the transport system, specifically, for estimating the effects of unavailable or eliminated edges, zones or transport modes [7].

The cusp catastrophe model created by Huang and his colleagues focuses on the analysis and the quantification of risks related to railway transport. It identifies the risk factors, the elements of the event chain, and the connections between them, using the bifurcation theory [8].

Hosseini and Verma deal with the optimization problem connected to the rail transportation of hazardous materials [9]. They aim to minimize the probability of accidents using a risk-based representation of the route choice problem.

However, these studies do not analyze in depth the possible interaction between an accident and the rescue operation connected to it. The present study focuses on this aspect, with special emphasis on the management of statistical methodological problems.

If a sudden catastrophe happens in a densely populated area, it is crucial to avoid congestions in the traffic system to ensure the appropriate coordination of mobility processes [10]. Without this, continuous traffic flow required for the rescue and emergency operations cannot be secured. Thus, areas affected by a catastrophe must be identified in order to be able to estimate the characteristics of congestions likely to occur. Using the Cusp Catastrophe Theory (CCT), Lin and his team suggest a special mathematical method to examine the possible traffic conditions of densely populated areas in the case of disasters [11].

According to Hu and his colleagues, rescue processes are influenced by several factors, including the time of the disaster, the intensity and place of secondary catastrophes, and the availability of rescue capacities. The authors created a multistep stochastic computer model to describe the disaster relief operations, taking into consideration different vehicle types, renting conditions, and public road conditions [12].

Gonçalves's team offers a comprehensive overview of resilience studies concerning urban transportation networks. The authors established a new framework for a complex survey of the resilience of transport systems in cities [13].

As the above review shows, the methods for the identification of the weak points in the public road system have been widely studied [14,15], but these studies primarily support real-time disaster scenes. Contrarily to these approaches, the present study aims to identify those elements in the network that can crucially reduce the resistance of the transport system. The methodology presented here allows for the reduction of disturbance



sensitivity of the network, which enhances the efficiency of the rescue teams in the case of a catastrophe.

Our hypothesis is that a new methodology can be prepared based on statistical and network optimization techniques. The availability of this method is introduced through the example of Budapest, although it is easy to set up for other settlements as well.

At first, the structure of the model is presented, which is followed by the description of the applied methods, and finally, the results are evaluated.

## 2. Boundaries and Structure of the Examined Network, Description of Scenarios

A key objective of the present research is to provide an objective, modern, and uniform methodology based on digital geographic information systems (GIS) for the yearly updated categorization of settlements for disaster management. Gradually, the new system is to replace the traditional, subjective methods.

Let us say we have a network that suffers a disaster that partially or completely destroys part of the infrastructure. The goal is to examine how the undesirable case affects the network's shortest path and maximum flow capacity between points and assumed zones. Since we are assuming a significant disaster, the traffic will likely be limited to emergency response vehicles; thus, dynamic traffic conditions can be dispensed with. The model is suitable for examining different scenarios that are assumed to arise due to the disaster. The maximum flow capacity and the shortest path can be compared between the tested points on the original network and the damaged network, as a result of which critical infrastructure elements can be identified.

This article presents a pilot-study for the above-described network research, focusing on the Hungarian capital, Budapest and its agglomeration area (Figure 1), which is a frequently researched area in transportation sciences, such as [16–20]. In general, Budapest is divided by the river Danube. The eastern side is referred to as the Pest side, while the western is called the Buda side, based on the historical settlements. The study area is divided into zones, which are set up along different conditions within and outside the capital. Concerning the agglomeration area, those settlements were selected which are served by suburban bus links or in which an intersection that is an integral part of the Budapest road network is present. In the agglomeration area, one settlement corresponds to one zone, with the exception of those settlement parts that are far from the central area of the settlement (e.g., Gyál-Némediszőlő (ID: 80)). Furthermore, large settlements were divided into smaller zones (e.g., Szigetszentmiklós (ID: 74–76), Érd (ID: 63–65)). As for Budapest, each district was cut up into 2–3 zones. Thus, the study area is made up of 96 zones altogether.

In correspondence with the methodological framework of the applied algorithms, the model of the public road network in the study area was created based on graph theory. It is important to highlight that the graph representation conforms to the algorithms and the test methods to be applied. The graph model of the study area is made up of 4922 edges and 4267 nodes.

In the pilot study, certain elements of the network were eliminated, with 100% reduction in their capacity. This models a major disaster, which causes longer travel times owing to the loss of graph elements. Furthermore, it is supposed that the catastrophe causes normal transport needs to cease, i.e., normal mobility needs are not taken into consideration. Roads are to be used solely for the rescue operations.

Three different scenarios were defined. Two factors determined which network elements should be deleted. On the one hand, network elements that are to be damaged with a probability higher than the average were selected for the different scenarios. Furthermore, real-life disasters (e.g., earthquakes, floods, accidents, terror attacks) might be associated to the selected components. On the other hand, elements exercising a major influence on the operation of the network were also selected.

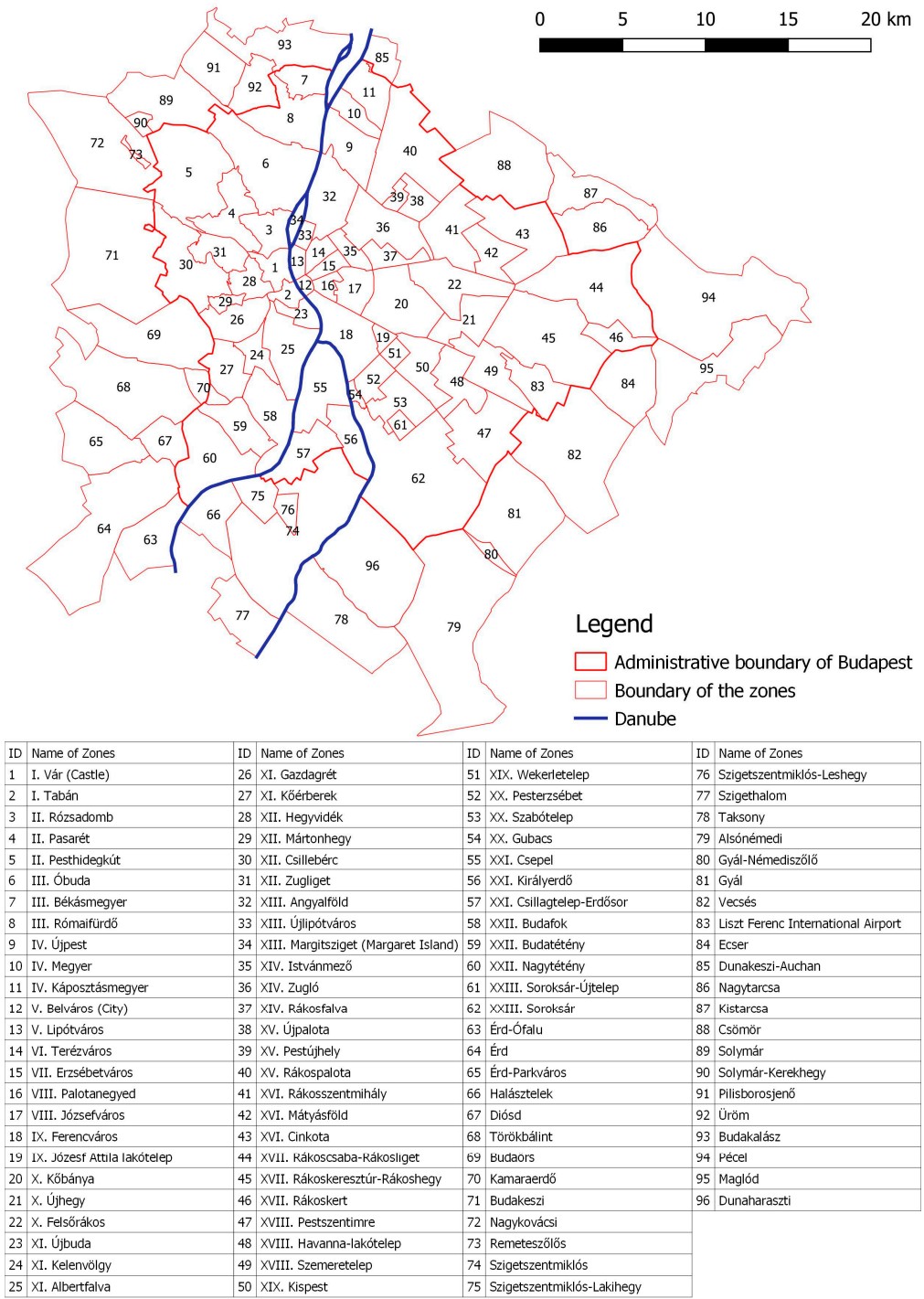

**Figure 1.** Zone names for the network (source: own work).

| ID | Name of Zones | ID | Name of Zones | ID | Name of Zones | ID | Name of Zones |
|---|---|---|---|---|---|---|---|
| 1 | I. Vár (Castle) | 26 | XI. Gazdagrét | 51 | XIX. Wekerletelep | 76 | Szigetszentmiklós-Leshegy |
| 2 | I. Tabán | 27 | XI. Kőérberek | 52 | XX. Pesterzsébet | 77 | Szigethalom |
| 3 | II. Rózsadomb | 28 | XII. Hegyvidék | 53 | XX. Szabótelep | 78 | Taksony |
| 4 | II. Pasarét | 29 | XII. Mártonhegy | 54 | XX. Gubacs | 79 | Alsónémedi |
| 5 | II. Pesthidegkút | 30 | XII. Csillebérc | 55 | XXI. Csepel | 80 | Gyál-Némediszőlő |
| 6 | III. Óbuda | 31 | XII. Zugliget | 56 | XXI. Királyerdő | 81 | Gyál |
| 7 | III. Békásmegyer | 32 | XIII. Angyalföld | 57 | XXI. Csillagtelep-Erdősor | 82 | Vecsés |
| 8 | III. Rómaifürdő | 33 | XIII. Újlipótváros | 58 | XXII. Budafok | 83 | Liszt Ferenc International Airport |
| 9 | IV. Újpest | 34 | XIII. Margitsziget (Margaret Island) | 59 | XXII. Budatétény | 84 | Ecser |
| 10 | IV. Megyer | 35 | XIV. Istvánmező | 60 | XXII. Nagytétény | 85 | Dunakeszi-Auchan |
| 11 | IV. Káposztásmegyer | 36 | XIV. Zugló | 61 | XXIII. Soroksár-Újtelep | 86 | Nagytarcsa |
| 12 | V. Belváros (City) | 37 | XIV. Rákosfalva | 62 | XXIII. Soroksár | 87 | Kistarcsa |
| 13 | V. Lipótváros | 38 | XV. Újpalota | 63 | Érd-Ófalu | 88 | Csömör |
| 14 | VI. Terézváros | 39 | XV. Pestújhely | 64 | Érd | 89 | Solymár |
| 15 | VII. Erzsébetváros | 40 | XV. Rákospalota | 65 | Érd-Parkváros | 90 | Solymár-Kerekhegy |
| 16 | VIII. Palotanegyed | 41 | XVI. Rákosszentmihály | 66 | Halásztelek | 91 | Pilisborosjenő |
| 17 | VIII. Józsefváros | 42 | XVI. Mátyásföld | 67 | Diósd | 92 | Üröm |
| 18 | IX. Ferencváros | 43 | XVI. Cinkota | 68 | Törökbálint | 93 | Budakalász |
| 19 | IX. József Attila lakótelep | 44 | XVII. Rákoscsaba-Rákosliget | 69 | Budaörs | 94 | Pécel |
| 20 | X. Kőbánya | 45 | XVII. Rákoskeresztúr-Rákoshegy | 70 | Kamaraerdő | 95 | Maglód |
| 21 | X. Újhegy | 46 | XVII. Rákoskert | 71 | Budakeszi | 96 | Dunaharaszti |
| 22 | X. Felsőrákos | 47 | XVIII. Pestszentimre | 72 | Nagykovácsi | | |
| 23 | XI. Újbuda | 48 | XVIII. Havanna-lakótelep | 73 | Remeteszőlős | | |
| 24 | XI. Kelenvölgy | 49 | XVIII. Szemeretelep | 74 | Szigetszentmiklós | | |
| 25 | XI. Albertfalva | 50 | XIX. Kispest | 75 | Szigetszentmiklós-Lakihegy | | |

## 2.1. Scenario A—Separation of the Public Road Network of Csepel

Csepel (ID: 55–57), which is District XXI of Budapest, is situated on the Csepel Island (Figure 2). The weakest point in the road infrastructure of the area is that there are very few links (i.e., bridges) leading to neighboring districts, and the capacity of the existing bridges is very low. As a result, traffic from the agglomeration area floods the residential areas of Csepel, and the bottleneck effect delimits the development of industrial areas [21].

Consequently, the number of links between the Csepel traffic system and Budapest, and between the district and the national road network is very limited. If the bridges are damaged, it exercises a critical effect on the accessibility of the island. Furthermore, if the

decrease in capacity occurs as a result of or together with an outer cause that endangers human lives, the decreased capacity also hinders escape and rescue operations and damage control. Owing to the severity of these consequences, the first surveyed scenario models the separation of the Csepel public road network.

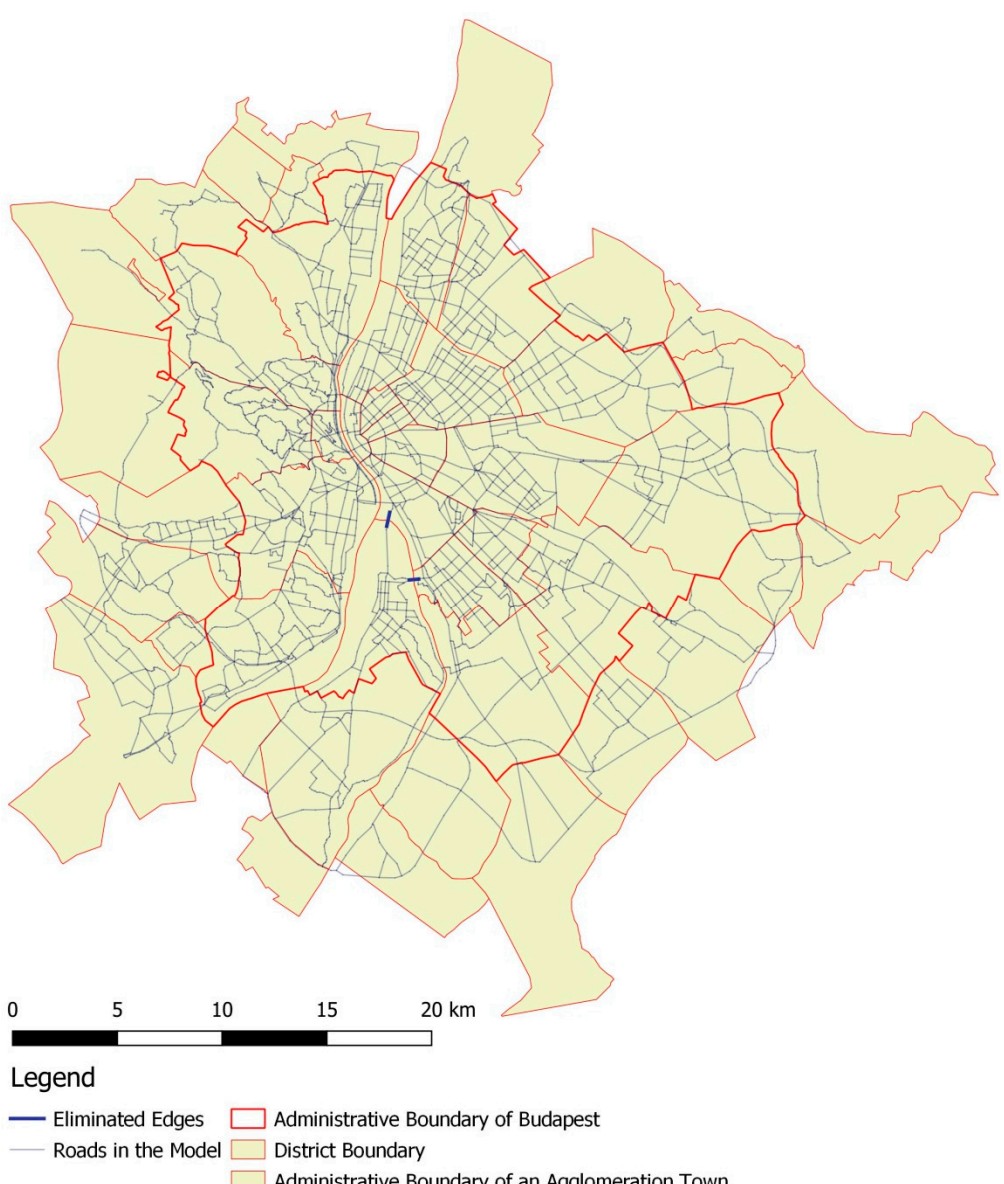

**Figure 2.** Scenario A—separation of the Csepel public road network (source: own work).

### 2.2. Scenario B—Separation of the South Buda Agglomeration Zone

Based on the practical experiences, the most critical characteristic of the road network in South Buda and the adjacent agglomeration zones is the lack of redundancy in the system connecting the area to the inner districts, and the low capacity of the existing linking roads (Figure 3). Moreover, the existing linking elements are situated close to the Kelenföld Railway Station, which enhances the risks of a possible disaster.

As a result of this topology, major congestions often occur in peak hours even during normal operation. Furthermore, a minor accident might also cause considerable traffic congestions.

As the number of main roads connecting the South Buda area with the inner city is rather restricted, their damage might have a critical effect on the accessibility of the area. If the capacity of these main roads is decreased as a result of or at the same time as a

dangerous, life-threatening event, this can make escape actions, rescue, and damage control processes more difficult. Therefore, Scenario 2 is the separation of the South Buda area from the inner city of Budapest.

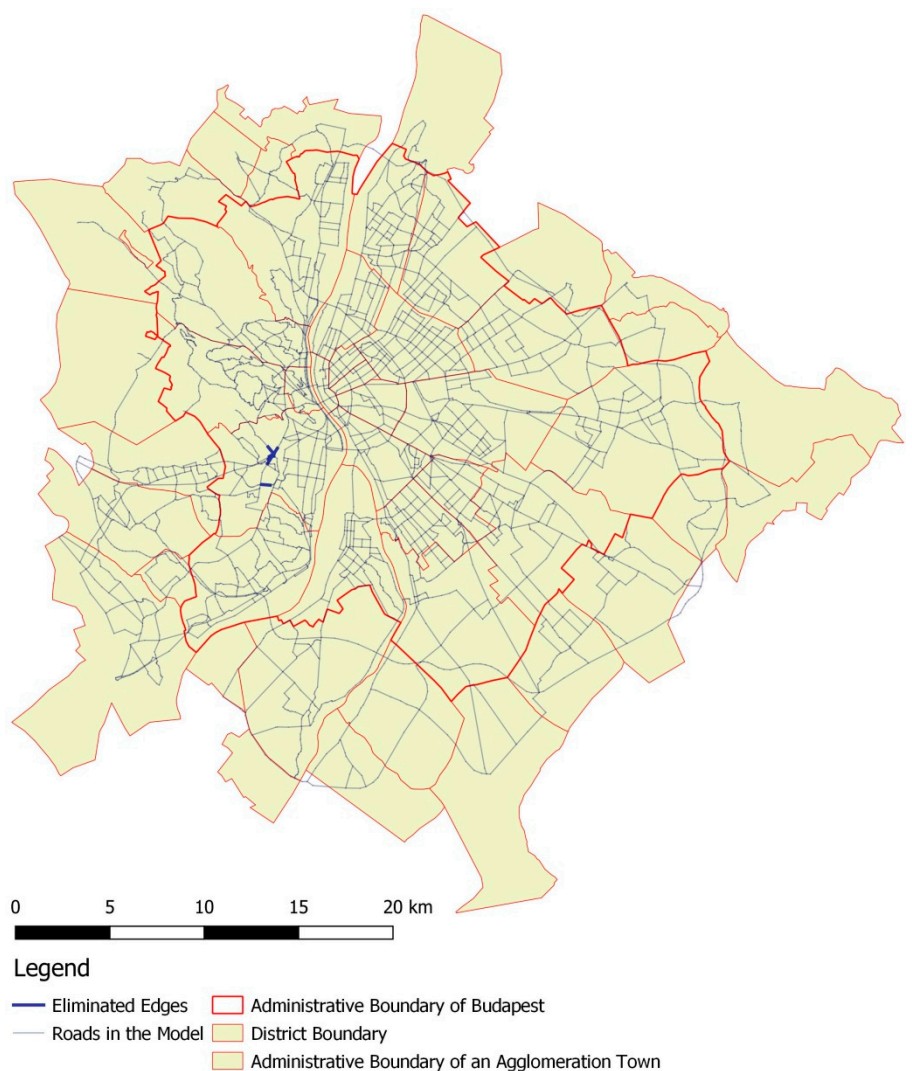

**Figure 3.** Scenario B—separation of the South Buda agglomeration zone (source: own work).

### 2.3. Scenario C—Damage of the Lower Quays

The roads running along the lower quays on both the Pest and Buda sides are integral parts of the road network of the capital and play a key role in satisfying the mobility needs (Figure 4). The quays of Budapest form a network that does not only serve the North–South through traffic, but also has a considerable role in satisfying local transport needs [21].

Consequently, it is evident that if the quays cannot function as normal, it affects the operation of the whole traffic network of the capital. If the capacity of these roads is lowered owing to hazards threatening human lives, the effect is more severe, as evacuation, rescue, and damage control processes might also be hindered. In reality, Budapest quays must be closed from time to time, as major floods block the roads. As supported by real-life cases, the surveying of this scenario is crucial.

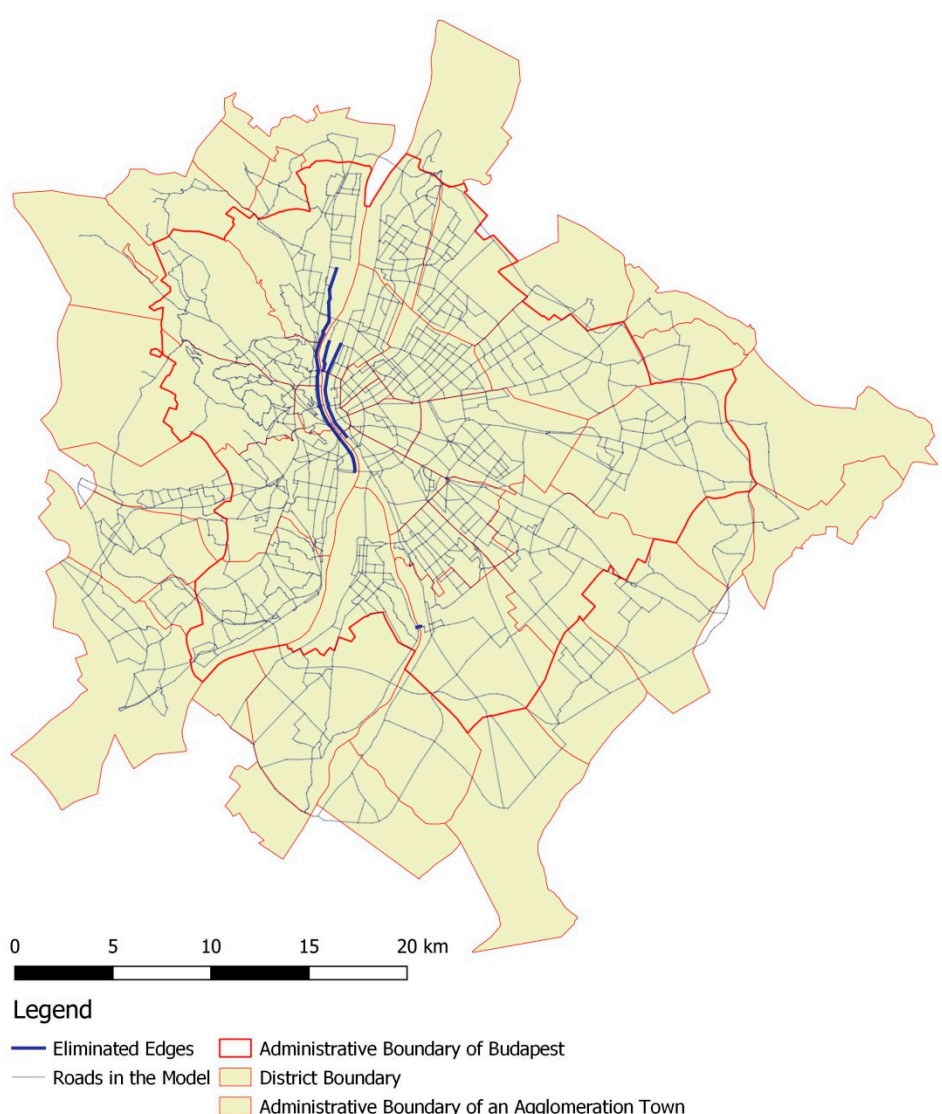

**Figure 4.** Scenario C—damage of the lower quays (source: own work).

## 3. Methods

The applied statistical and network optimization techniques are presented in the methodological part. The goal of this article was not to compare the different methodologies or shed light on the differences in efficiency. With the current IT development level, the network can be considered small, so using other methods does not significantly affect the running time. Of course, a future research question may be whether the network optimization procedures used here are still adequate for significantly larger networks.

### 3.1. Analysis of Travel Distance Changes

In the first step of analysis, it is examined how the damage of a given network element influences the accessibility of a given point in the system from another point. The major reason for this is that in the case of a disaster, needs for rescue operations, damage control, and restoration processes override normal traffic needs.

In order to reduce the calculation capacity required, the present study examines the travel distance changes between major junction nodes and zone center nodes. The distances between the examined nodes were estimated by Dijkstra's algorithm [22].

Let us suppose the graph $G = (V, E, s, c)$, where $V = (v_0, \ldots, v_{N-1})$ is the set of nodes in the graph; $E \subseteq V^2$ is the set of edges in the graph; $s \subseteq V$ is the starting point for the

given trip; and weights representing the sections is marked by $c : E \rightarrow \mathbb{N}$ [23]. The logical function $B_n$ is responsible for marking the relevant edges, and the function is defined as $f : \{0, 1\}^n \rightarrow \{0, 1\}$. Let the *i*th element of the binary assignment be $x \in \{0, 1\}^n$, marked by the indexed parameter $x_i$, and let its value be $|x| := \sum_{i=0}^{n-1} x_i 2^i$. In harmony with the above, *G* can be defined with the relation $\chi E \in B_{2n}$, which gives the binary pairs of nodes $(x, y) \in \{0,1\}^{2n}$ if $\left( v_{|x|}, v_{|y|} \right) \in E$ and $c\left( v_{|x|}, v_{|y|} \right) = |d|$. The algorithm uses the functions $C(x, y, d)$ and $s(x)$ as inputs.

$$C(x, y, d) = 1 \Leftrightarrow \left[ \left( v_{|x|}, v_{|y|} \right) \in E \right] \wedge \left[ c\left( v_{|x|}, v_{|y|} \right) = |d| \right], \tag{1}$$

$$s(x) = 1 \Leftrightarrow v_{|x|} = s. \tag{2}$$

The output function of the algorithm can be defined as $\text{dist} : V \rightarrow \mathbb{N}_0 \cup \{\infty\}$, which assigns the length of the shortest route between nodes $v$ and $s$ to $v \epsilon V$.

$$DIST(x, d) = 1 \Leftrightarrow dist\left( v_{|x|} \right) = |d|. \tag{3}$$

The above method described by symbols was implemented in the MATLAB software [24]. The program solves Dijkstra's algorithm for $l \cdot m \cdot (m - 1)/2$ cases, where *m* is the number of nodes serving as zone centers and *l* is the number of nodes eliminated independently from the examined graph.

During the method, the currently examined points are divided into two parts. The permanent list contains the points already examined, and the tentative list includes their neighbors. Along the procedure, the element in the tentative list with the smallest distance is transferred to the permanent list. Then, the adjacent elements not yet examined are added to the tentative list [25]. The flowchart of the algorithm can be seen in Figure 5.

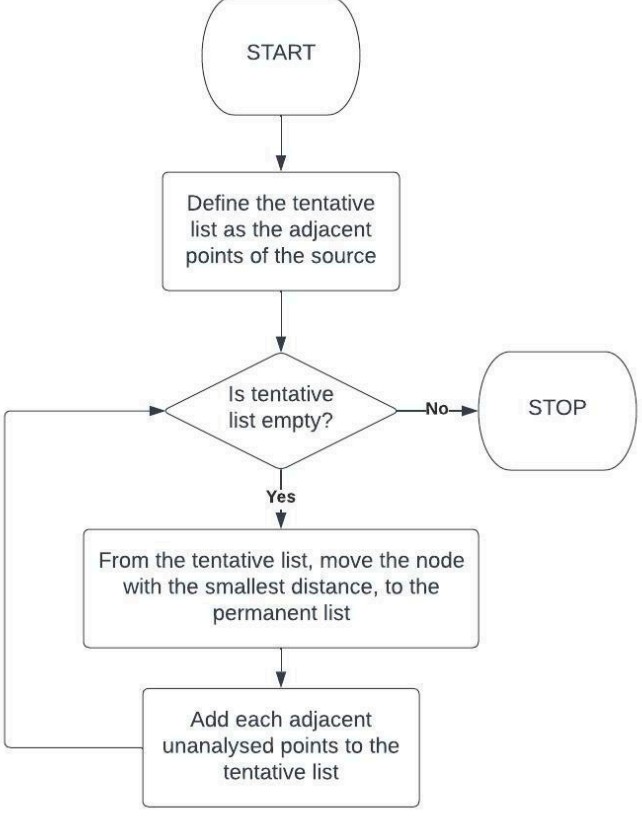

**Figure 5.** Flowchart of Dijkstra's algorithm (source: own edition, based upon [25]).

### 3.2. Analysis of Capacity Change

The second step of the analysis is carried out to identify the public road network elements critical for damage control and rescue operations. To define the maximum rescue capacity, the solution of the maximum flow problem was applied. This can be simply explained as the identification of the upper boundary of the amount of a specific fluid that can flow through a pipeline. Based on the results of Ford and Fulkerson [26], it can be stated that maximum flow and minimum cut problems are equivalent. Boykov and Kolmogorov [27] introduced a novel method to increase the efficiency of standard augmenting route methods. The flowchart of the method can be seen in Figure 6.

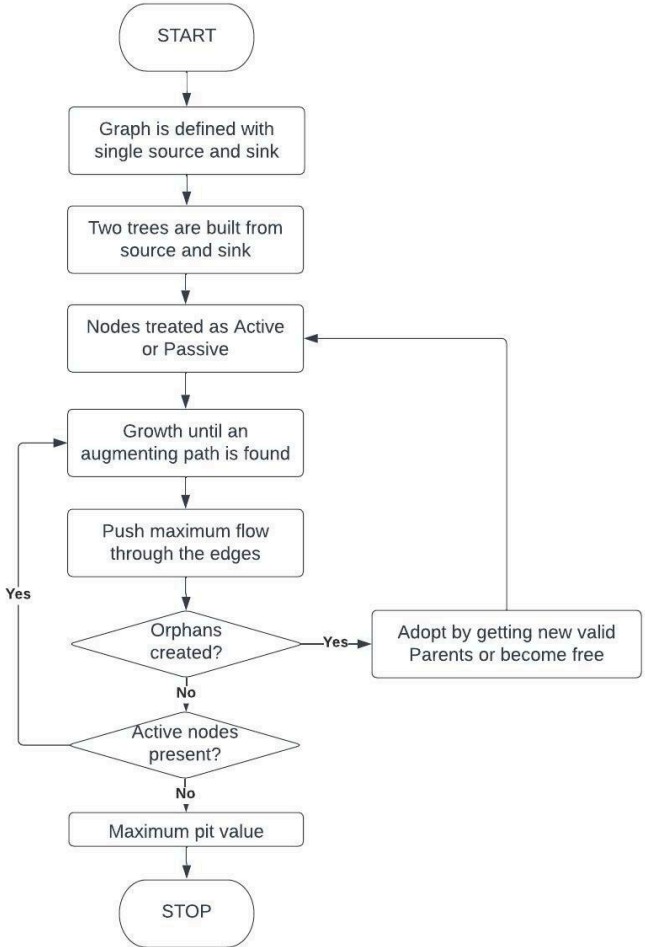

**Figure 6.** Flowchart of Boykov–Kolmogorov algorithm (source: own edition, based upon [28]).

According to the Boykov–Kolmogorov approach, during the growth stage, child nodes are assigned to the selected network components. The augmentation phase needs a feasible R route between the initial and the ending node. At the beginning of this phase, the set of components without parents is empty. However, finally, there can some components in this set, since the capacity of at least one link from R will be exhausted. In the adoption stage, network components without parents are assigned to a parent until the set of components without parents is emptied.

The above-described method is implemented in the maxflow function of MATLAB. This function was chosen because other built-in algorithms (e.g., the Ford–Fulkerson algorithm) cannot be used for undirected graphs [29]. The program maps the Boykov–Kolmogorov algorithm for a number of $lm(m-1)/2$ cases, where $m$ is the number of nodes serving as zone centers and $l$ is the number of examined scenarios.

### 3.3. Comparison of Scenarios

Statistical methods were used to compare scenarios A, B, and C. The necessary normality test for the samples (the shortest routes [m] leading from all other zones from a given zone, for different scenarios) was carried out with the Shapiro–Wilk test [30]. The Friedman test [31] was chosen from the non-parametric tests, as our samples are not independent, but paired.

During the Friedman test, $m$ paired samples are compared in order to determine whether they were taken from the same distribution. An $m \times n$ matrix ($X^{m \times n}$) can be created from the vectors of the paired samples, where $n$ is the number of observations. Then, the $R^{m \times n}$ matrix is created from the observation matrix, whose elements ($r_{ij}$) give where the $i$ value that belongs to the given sample $j$ is ranked out of the $i$ values of all samples, while the following two conditions must be met:

$$1 < r_{ij} < m \ \forall i, j, \tag{4}$$

$$\sum_{i=1}^{m} r_{ij} = \sum_{k=1}^{m} k. \tag{5}$$

The following equation is used to determine the value of the Friedman test statistics (6). The test statistic value has a $\chi^2$-distribution with an $m - 1$ degree of freedom.

$$F = \frac{12}{nm(m+1)} \sum_{i=1}^{m} R_i^2 - 3n(m+1) \sim \chi^2_{m-1}, \tag{6}$$

where:

$$R_i = \sum_{j=1}^{n} r_{ij} \ \forall i = 1 \dots m. \tag{7}$$

Without increasing the Family-wise Error Rate (FWER), there are two generally accepted approaches for a pairwise comparison of cases matched in the Friedman test: post hoc tests and contrast tests. As only weak hypotheses exist for the results of our surveys, i.e., we cannot estimate which scenarios will be different, a post hoc test is required. Therefore, more strict requirements are set up for significancy. After proving the homogeneity of standard errors, the Bonferroni correction [32,33] was applied.

As the samples did not have a normal distribution, it is advisable to apply a non-parametric test for post hoc testing, i.e., the Wilcoxon test [34]. The test is applied to examine the homogeneity of matched samples ($x$ and $y$). During this test, the absolute difference between the elements of two samples $|x_i - y_i \ \forall i|$ are assigned a rank $r_i$, according to their place in the ordered set of absolute differences. The value of the test statistic is determined with Equation (8).

$$R_+ = \sum_{i=1}^{n} r_{i|x_i-y_i>0 \ \forall i} \sim \mathcal{N}(\mu_+, \sigma_+), \tag{8}$$

where:

- $\mu_+ = \frac{n(n+1)}{4}$;
- $\sigma_+ = \sqrt{\frac{n(n+1)(2n+1)}{24}}$;
- $n$: number of sample elements.

The significance values ($p$ values) of the Wilcoxon tests are modified by the Bonferroni correction, in order to eliminate the FWER. Therefore, it is not the original value of $p$ that is tested whether it is in the expected confidence interval, but the modified $p$ value (9) [33].

$$\text{Modified } p = p \times \frac{m(m+1)}{2} \tag{9}$$

where $m$ is the number of samples undergoing the Friedman test.

## 4. Results

*4.1. Analysis of the Scenarios*

The damaging of the network was modelled by eliminating certain edges $\left(w_{|x|}, w_{|y|}\right) \in E$ from the graph. The analysis considered each evaluated pair of nodes $\left(w_{|x|}, w_{|y|}\right)$, and for each analysis the set of examined edges is defined as $E \backslash \left(w_{|x|}, w_{|y|}\right)$, $\left(w_{|x|}, w_{|y|} \in E\right)$. All the nodes considered as zone centers were examined in each graph where a given pair of nodes $w_{|x|}, w_{|y|}$ were eliminated from the graph.

### 4.1.1. Analysis of Scenario A

Because Csepel is situated on an island and has very few connections to the neighboring zones, it was supposed that the two bridges within the capital would be eliminated from the public road system. This would force the rescue teams to make considerably longer trips to reach the island.

On the basis of the ranking of relations between Csepel and other model zones, it was proven that highest values characterize the relations between Csepel and North Buda (estimated shortest extra route: ~11 km) and Csepel and Northeast Pest (estimated shortest extra route: ~10 km).

Based on the summed values, the set of zone central nodes can be determined which are most affected by the elimination of bridges to the island. All such nodes are situated within the district of Csepel: Csepel-Center (ID: 55) (sum of shortest estimated extra routes: ~641 km) and Csepel-Erdősor (ID: 57) (sum of shortest estimated extra routes: ~ 387 km). Relatively high values were attested for some other nodes in the region, such as Szigetszentmiklós-Center (ID: 74) (sum of shortest estimated extra routes: ~88 km), Lakihegy (ID: 75) (sum of shortest estimated extra routes: ~67 km), and Leshegy (ID: 76) (sum of shortest estimated extra routes: ~174 km).

The length of extra routes is not significant for Szigethalom (ID: 77), as the shortest routes lead across the Taksony vezér Bridge to the Pest side and towards the Buda side, across the Deák Ferenc Bridge, which were not eliminated in the model. Furthermore, the Soroksár Ferry is also relatively close. However, it must be noted that these elements of the public road network have low capacity, which must be considered for rescue operations in the case of a disaster.

By the comparison of the decrease in capacity owing to the damaging of certain elements of the road network, the relations between the island and the neighboring districts can be ranked. As a result of this process, several relations with high absolute values were identified. The highest values were recorded for relations between Csepel district and the nodes in the inner city and Northeast Pest (highest estimated capacity decrease in lane-number-unit: −2).

Based on summed values, the nodes serving as zone centers affected most by the elimination of bridges can be identified. The highest absolute value was assigned to the Csepel-Center node (highest summed estimated capacity decrease in lane-number-unit: −45).

### 4.1.2. Analysis of Scenario B

In this scenario, those elements of the road network were eliminated that link the South Buda area with the central area of the capital. Owing to this damage, rescue teams can reach the area via considerably longer routes.

The analysis of the relationship between south Buda and the inner zones proved that the highest values were attested for the pair South Buda agglomeration and Újbuda (ID: 23) (estimated shortest extra route: ~3 km), and the South Buda agglomeration and the Inner City (ID: 12) (estimated shortest extra route: ~3 km).

Based on the summed values, the nodes most effected by the elimination of edges in this scenario were selected. The nodes with the highest values are Kőérberek (ID: 27) (sum of shortest estimated extra routes: ~138 km) and Érd-Parkváros (ID: 65) in the agglomeration

(sum of shortest estimated extra routes: ~133 km). High values were also calculated for Törökbálint (ID: 68) (sum of shortest estimated extra routes: ~127 km), Budaörs (ID: 69) (sum of shortest estimated extra routes: ~123 km) zones and within the capital, Kamaraerdő (ID: 70) (sum of shortest estimated extra routes: ~130 km). In the case of Érd-Ófalu (ID: 63), the increase is not significant, as other main roads (e.g., main road no. 6 along the Danube) are good alternatives to approach the inner city of Budapest. However, the capacity of these alternative roads is much lower than that of the eliminated parallel expressways, i.e., this can cause severe difficulties in the case of a disaster.

Based on the capacity decrease arising when edges are eliminated, the relations between the examined area and the city center were ranked. Interestingly, the elimination of the chosen edges does not reduce the capacity of the road network considerably. The reason for this is that the capacity of parallel routes proved to be at least as much as that of the eliminated edges. Consequently, the damage of the chosen edges would not hinder escaping or rescuing considerably. This means that our hypothesis formed on the basis of practical experience was not confirmed by the simulation. Thus, the sections eliminated in this scenario, i.e., roads connecting the South Buda area and the agglomeration with the inner area of Budapest, cannot be regarded as the most critical elements of the Budapest road network.

### 4.1.3. Analysis of Scenario C

This scenario showed what would happen if both quays, which connect the northern and southern parts of the city, were damaged, for example, due to a major flood. This would mean that rescue teams would have to cover longer distances to reach their destinations.

According to the ranking of the relations between the nodes after the elimination of both quays, no outstanding values were found. This may be due to the fact that parallel routes have approximately the same lengths. It must be noted that the Margit Island, which is accessible from two bridges, might be difficult to access, as the routes from the quays to the relevant bridges cannot be used. Therefore, alternative routes must be taken to access these bridges, which may increase the length of access routes. Furthermore, it is advisable to consider the possibility that the Soroksár–Csepel Ferry stops operation. Although this fact would not significantly influence the rescue operations, it might significantly lengthen the shortest path between South Csepel and South Pest.

Relations most sensitive to the damage of the north–south axis were ranked based on the capacity reduction. No values with outstandingly high absolute values were found. This may be due to the fact that if the quays are closed, low capacities would not emerge in the alternative parallel routes, but rather in the elements of the local road network.

The zone center nodes most affected by the elimination of the quays proved to be Pilisborosjenő (ID: 91) in the agglomeration (highest summed estimated capacity decrease in lane-number-unit: −180).

### 4.2. The Effects of the Scenarios on the Whole Network

The road network examined here was divided into 96 zones. For each scenario, it was examined how the shortest routes change between the zones. At first it was examined whether the shortest routes from a given zone have a normal distribution. Because in several cases this was not proven, only non-parametric statistical tests could be applied (i.e., Wilcoxon and Friedman). If one zone is found where the assumption of the normal distribution cannot be accepted, only non-parametric tests (Wilcoxon and Friedman) can be used for statistical analysis to make the results comparable.

To decide whether $t$-test and ANOVA can be used instead of non-parametric tests, the Shapiro–Wilk test is used for normality. The data analyzed for each and every zone is the shortest route from the given area to every other ($n = 95$). For example, let us consider the Óbuda zone (ID: 6), the lightest grey zone (in Figure 7) in the northwestern part of the city. The shortest route to every other zone is collected, and the Shapiro–Wilk test is carried out on the data. The significance value of the test is between 0 and 0.001; thus, the hypothesis

of normal distribution for the analyzed shortest route data must be rejected. This means the shortest routes from the given zone to every other zone do not follow normal distribution. The results of the normality tests for each zone are shown in Figure 7.

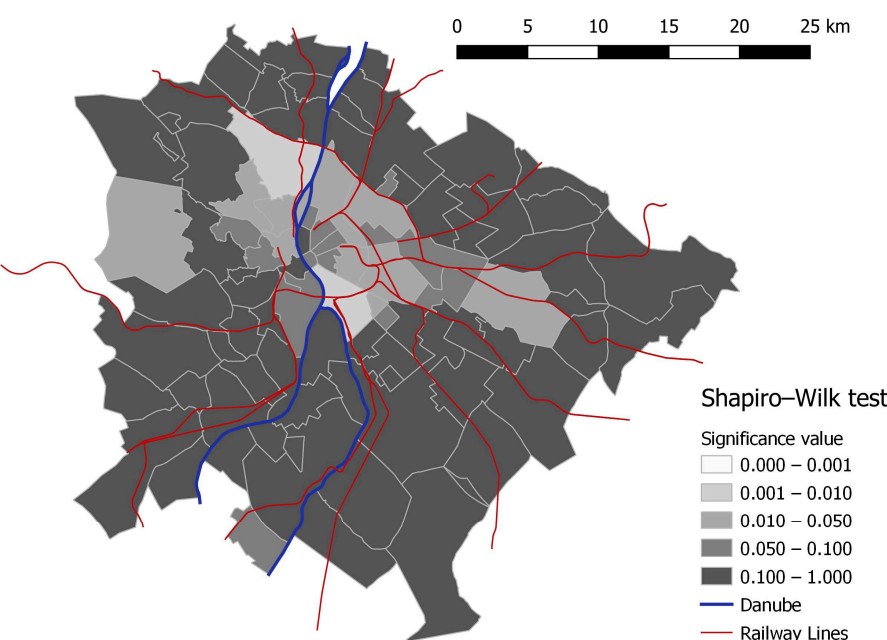

**Figure 7.** Results of the normality test for each zone (source: own work).

The shortest routes do not follow a normal distribution in the following cases: (i) the routes from the designated central node are restricted, owing to the hydrological, terrain or city structure of the area; e.g., the Caste District (Várkerület) (ID: 1), Margit-sziget (ID: 34), Rákoshegy (ID: 45), Budakeszi (ID: 71) [35]; (ii) a topographical separation exercises influence on the area, e.g., for Ferencváros (ID: 18), Óbuda (ID: 6) or Angyalföld (ID: 32), where the outer bridges over the Danube are found, or the outer reaches of the railway lines, where the tracks can be crossed at a limited number of crossings compared to inner areas, e.g., József Attila-lakótelep (ID: 19) and Kőbánya (ID: 20) along railway line 100, Józsefváros (ID: 17) along railway line 80/120, or Zugló (ID: 36) and Istvánmező (ID: 35) along railway line 70.

As the shortest routes from each zone were analyzed separately, each zone has a corresponding result of the Friedman test (Figure 8).

The examined scenarios affect the whole city, as the results of the Friedman test showed significant changes to most of the zones. The exceptions are some areas in the outskirts of northern and southern Pest. In the north, the road system is so dense in Pestújhely (ID: 39) and Rákospalota (ID: 40), that even if an element of the network is eliminated, no considerable changes occur in the shortest routes. Concerning South Pest (Pestszentimre (ID: 47), Havanna-lakótelep (ID: 48), Kispest (ID: 50)), the relative closeness of two major roads (Nagykőrösi út and Üllői út) might account for these results.

For each scenario (A, B, C), the Wilcoxon post hoc test [34] was carried out with Bonferroni correction [32,33]. Below, the results are presented.

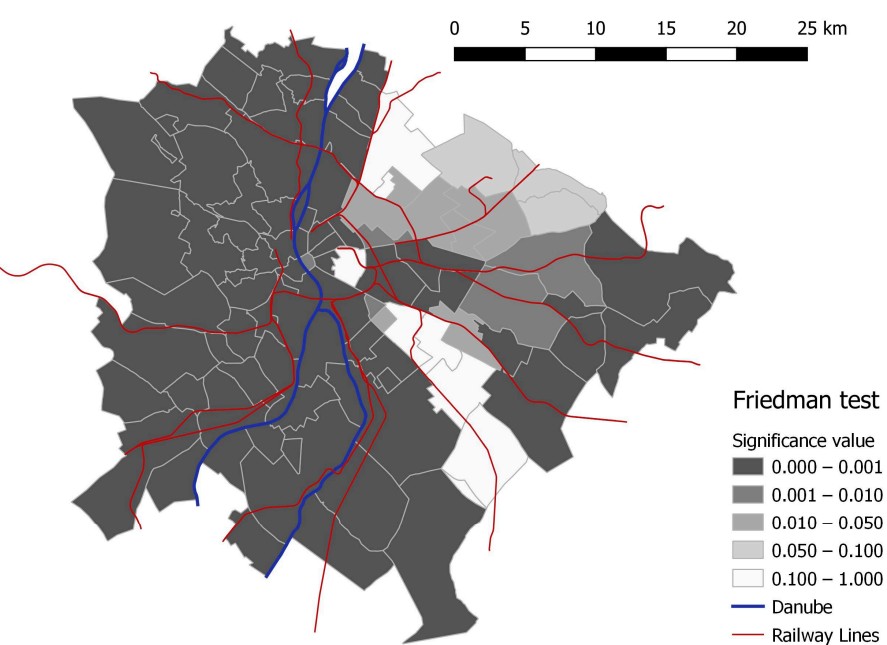

**Figure 8.** Results of the Friedman test for each zone (source: own work).

4.2.1. Scenario A

It was tested whether the shortest routes from a given zone to all other zones show the same distribution in normal circumstances and in Scenario A. If the bridges connecting Csepel to the rest of the city within the city boundary are eliminated, the shortest paths to other zones grow significantly only for zones in the Csepel Island. Such a closure would leave the rest of the city unaffected (Figure 9).

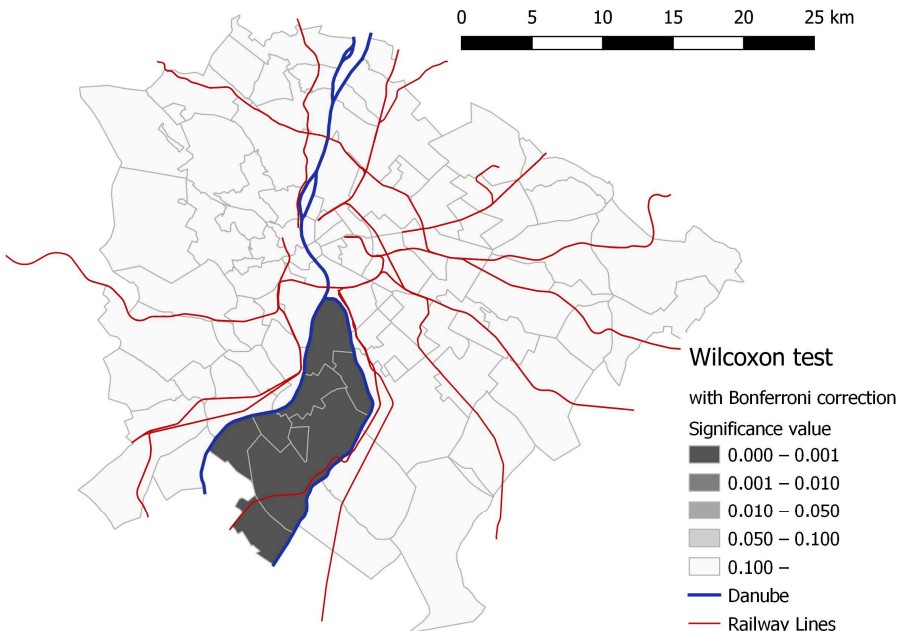

**Figure 9.** Effect of Scenario A, the result of the Wilcoxon test with Bonferroni correction (source: own work).

Consequently, it can be stated that the integration of the Csepel Island is not satisfactory at the moment. Some improvement is offered by the planned Galvani and Albertfalvi

Bridges toward the Buda side of the city. The accessibility of the island must be improved so that its vulnerability and the consequences of that can be reduced.

### 4.2.2. Scenario B

The post hoc test was carried out for the scenario modelling the elimination of the most important southwestern intersection (namely the mutual section of M1 and M7 expressways and the Egér Road). The results show that these can be considered as the most important edges in the South Buda region (Figure 10).

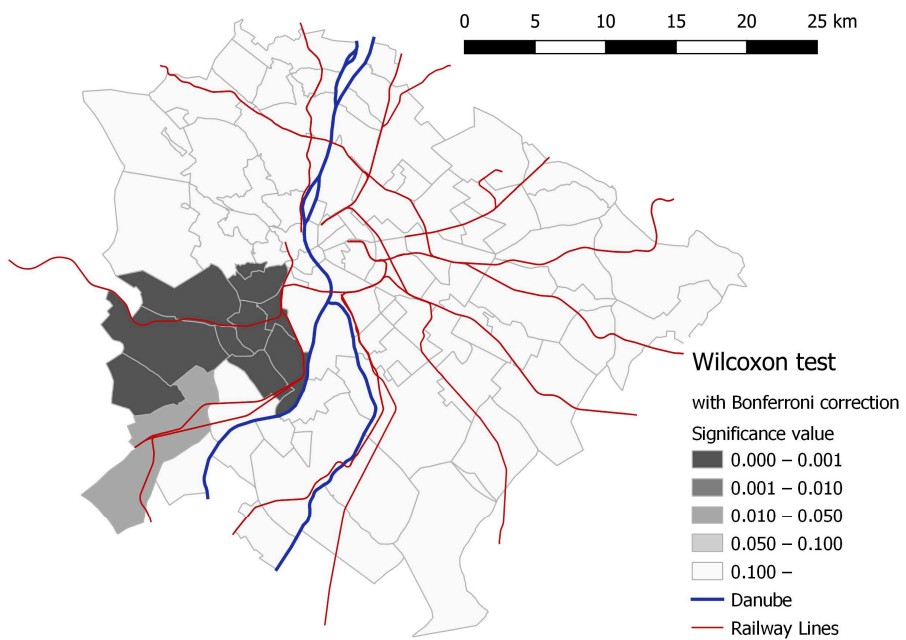

**Figure 10.** Effect of Scenario B, the result of the Wilcoxon test with Bonferroni correction (source: own work).

As mentioned before, this kind of elimination highly affects the southwestern area. However, as we take into consideration the zones closer to the Danube, the level of the Wilcoxon post hoc test seemed to be lower. This indicates the importance of the alternative main roads along the Danube (e.g., main road 6). Therefore, Érd-Ófalu (ID: 63) and Nagytétény (ID: 60) are not affected considerably, though to the north of these zones, the elimination of the network elements exercises its effects. The sensitivity of the system to problems might be lessened if the western sector of ring road M0 was completed, and the construction of the earlier planned outer road might also improve the circumstances.

### 4.2.3. Scenario C

The post hoc test was carried out for the flood scenario as well. If both quays are eliminated, the Margit Island and the zones along the Danube in the outskirts are affected the most (Figure 11).

In the southern regions, the effects are enhanced by the supposed closure of the D14 Ferry (Soroksár–Csepel). The inclusion of this element of the system might seem problematic, as this ferry cannot be regarded as an element of full value. However, at the local level, it plays an important role. Consequently, the development of the ferry or its replacement with a bridge should be considered. In the inner areas, the closure of the quays does not cause a significant change in route lengths, as the road system is dense, so the number of detour routes is high.

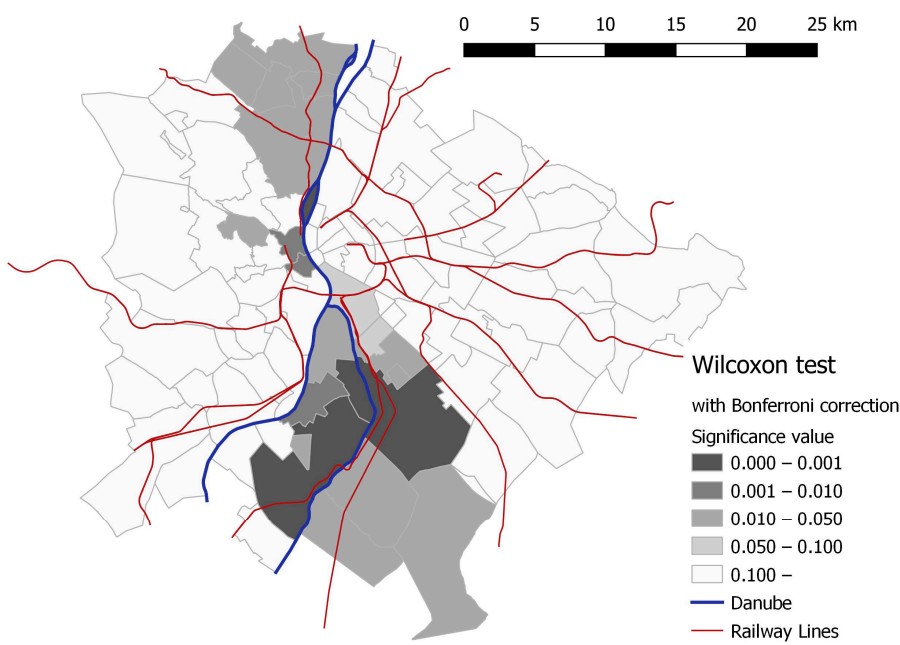

**Figure 11.** Effect of Scenario C, the result of the Wilcoxon test with Bonferroni correction (source: own work).

## 5. Conclusions

This study describes a method for the analysis of those components of the public road system that can critically affect the operation of the system in the case of a disaster, which can be crucial when defining the basis of city planning [36,37]. The road network of the pilot study area (Budapest and its agglomeration) was modelled as a graph. Dijkstra's algorithm was applied to find the shortest path and the Boykov–Kolmogorov method was used to determine that the maximal flow were applied. As a result, the critical elements of the road network were identified, which would crucially hinder escaping, rescue operations, and damage control.

Three scenarios were examined, in which crucial elements of the road network were eliminated. These elements are more sensitive to damage than the average. In Scenario A, the bridges leading to the Csepel Island within the city boundary were deleted, as a result of which the public road network of Csepel was separated from the rest of the city. Scenario B modelled the separation of the South Buda region, while Scenario C simulated the effect of a flood blocking both quays along the Danube.

During the examination of each scenario, relations and nodes critical for travel distance and rescue capacities were identified. This method thus can be used to verify the strategy of damage control. The paired samples were tested with the Friedman test to reveal whether the examined elements belong to the same distribution. This homogeneity test showed whether the different scenarios differ significantly, and whether a given closure exercises significant effects on a given zone. In the last step of the analysis, the effect of the elimination of the given elements was examined with the Wilcoxon post hoc test with Bonferroni correction.

Concerning the limitations of this study, the mapping of available capacities during a rescue operation and the traffic conditions must be mentioned. The present study hypothesizes that during a rescue operation, the normal traffic is not present on the road network. This hypothesis may be considered as valid in a considerable number of disastrous events. Our basic hypothesis was that in the case of a life-threatening disaster which poses a continuous threat (e.g., strong earthquake, accident resulting in significant radioactive radiation, multiple terror attacks, etc.), the traffic network cannot be characterized by the normal traffic conditions. Mobility motivations arising in normal circumstances (such as work, education, shopping, and entertainment) are minimized. However, the efficiency of

estimation can clearly be improved if the normal traffic is also taken into consideration for the infrastructure capacity calculations.

This limitation sets the directions for further research. If the traffic model is integrated, the disaster estimation model can become more effective.

Methods applied for the analysis of the effects of system element damage can be used in a wide range of fields and can be adapted to other networks as well. The methods chosen for this study can be applied even is special conditions concerning sample characteristics (e.g., normality) are not met.

**Author Contributions:** T.S.: conceptualization, methodology, statistical analysis, evaluation, writing and editing. Z.S.: methodology, statistical analysis. M.O.: methodology, graph-based investigation, validation. Á.T.: methodology, graph-based investigation, coordination, review. All authors have read and agreed to the published version of the manuscript.

**Funding:** This research was supported by the Ministry of Innovation and Technology NRDI Office within the framework of the Autonomous Systems National Laboratory Program.

**Institutional Review Board Statement:** Not applicable.

**Informed Consent Statement:** Not applicable.

**Data Availability Statement:** The datasets used and/or analyzed during the current study are available from the corresponding author on reasonable request.

**Acknowledgments:** The authors are grateful for the support of the New National Excellence Programme Bolyai+ scholarship.

**Conflicts of Interest:** The authors declare no conflict of interest.

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
