# Peer review of "Disaster Risk Assessment Scheme—A Road System Survey for Budapest"

_sustainability, doi:10.3390/su15086777_

Round 1

Reviewer 1 Report

Dear authors of the article, 

The article is devoted to the actual task of studying the transport infrastructure of the city (based on  on Budapest) on the basis of graph theory, the search for weaknesses in the transport infrastructure. The use of models and methods, algorithms of graph theory allows us to solve the tasks. The use of the MatLab software system allows finding a solution to a complex transport problem.

The choice of specific scenarios indicates a serious study of the infrastructure of the city of Budapest.

The results of the study are presented in detail in the figures (Figures 3-8).

However, in the article, the authors do not point out the dynamic features of traffic flows, irregularities during the day, and features of the organization of traffic. Introducing the term disaster, it is desirable to stay on a specific one scenario. 

There are typos in several places in article that need to be corrected.

The article is constructed logically correctly. The list of references is up-to-date. The article solves an urgent problem in the field of research of the transport infrastructure of cities in cases of emergency, the results can be used to solve the forecasting of the development and modernization of the infrastructure of the transport networks of the city.   Based on article, the presented models and methods can be applied to other cities.

Reviewer 2 Report

The referee's statement has been uploaded for the article.

Round 2

Reviewer 2 Report

The article can be published as it is.